# Narrative Review of New Insight into the Influence of the COVID-19 Pandemic on Cardiovascular Care

**DOI:** 10.3390/medicina58111554

**Published:** 2022-10-29

**Authors:** Any Axelerad, Alina Zorina Stuparu, Lavinia Florenta Muja, Silviu Docu Axelerad, Silvia Georgeta Petrov, Anca Elena Gogu, Dragos Catalin Jianu

**Affiliations:** 1Department of Neurology, General Medicine Faculty, ‘Ovidius’ University, 900470 Constanta, Romania; 2Department of Neurology, ‘Sf. Ap. Andrei’ County Clinical Emergency Hospital of Constanta, 900591 Constanta, Romania; 3Faculty of General Medicine, ‘Vasile Goldis’ University, 317046 Arad, Romania; 4Doctoral School of the Faculty of Psychology and Educational Sciences within the University of Bucharest, 050663 Bucharest, Romania; 5Department of Neurology, “Victor Babeș” University of Medicine and Pharmacy, 300041 Timișoara, Romania; 6Centre for Cognitive Research in Neuropsychiatric Pathology (Neuropsy-Cog), Faculty of Medicine, “Victor Babeș” University of Medicine and Pharmacy, 300041 Timișoara, Romania

**Keywords:** cardiothoracic surgery, vascular surgery, COVID-19, pandemics, residency, surgeries

## Abstract

*Background and Objectives*: The purpose of this paper was to perform a literature review on the effects of the COVID-19 pandemic on cardiothoracic and vascular surgery care and departments. *Materials and Methods*: To conduct this evaluation, an electronic search of many databases was conducted, and the resulting papers were chosen and evaluated. *Results*: Firstly, we have addressed the impact of COVID-19 infection on the cardiovascular system from the pathophysiological and treatment points of view. Afterwards, we analyzed every cardiovascular disease that seemed to appear after a COVID-19 infection, emphasizing the treatment. In addition, we have analyzed the impact of the pandemic on the cardiothoracic and vascular departments in different countries and the transitions that appeared. Finally, we discussed the implications of the cardiothoracic and vascular specialists’ and residents’ work and studies on the pandemic. *Conclusions*: The global pandemic caused by SARS-CoV-2 compelled the vascular profession to review the treatment of certain vascular illnesses and find solutions to address the vascular consequences of COVID-19 infection. The collaboration between vascular surgeons, public health specialists, and epidemiologists must continue to investigate the impact of the pandemic and the response to the public health issue.

## 1. Introduction

The coronavirus disease-2019 (COVID-19) pandemic has caused widespread illness and death on a worldwide scale [1,2,3]. In accordance with public health guidelines on physical separation and to help save or allocate vital resources, regular medical treatment has been subject to reasonable, but considerable, constraints. The vast majority of invasive cardiovascular (CV) surgeries and diagnostic testing have been postponed, with cardiovascular associations urging for enhanced prioritization and treatment of patients on waiting lists [4]. Consequently, people with untreated cardiovascular disease are more likely to have unfavorable consequences [5].

## 2. Materials and Methods

Our investigation used the electronic scientific resources PubMed, Google Scholar, Web of Science, and Science Direct. Between 2019 and 2022, relevant English articles employing the phrases “cardiothoracic surgery”, “vascular surgery”, and “cardiothoracic and vascular surgery” in connection with “COVID-19” and “COVID-19 pandemic” were identified. The selection of 80 articles was based on database searches. The search was augmented manually with references from recent research and case reports. The correctness of the document titles was verified, and duplicates were eliminated. In all, 130 articles were discovered; 31 were omitted following full-text screening, and 20 were omitted due to duplication. The qualifying requirements were satisfied by scholarly literature on the impact of COVID-19 on departments of Cardiothoracic and Vascular Surgery.

The authors manually checked the reference lists of the selected literature for completeness in order to validate the inclusion of information pertinent to the effect of COVID-19 on departments of Cardiothoracic and Vascular Surgery. Articles were included in this review if they matched the following eligibility requirements: (1) they were published in English, and (2) they were original research articles, reviews, or case reports. Studies that satisfied the following criteria were excluded: (1) citations and patents, (2) authored in a language other than English, and (3) abstract articles with no data.

## 3. Context

Patients with documented cardiovascular disease will always be impacted by treatment delays. Furthermore, decreased accessibility to diagnostic screening will result in a high prevalence of undetected cardiovascular illness, thus delaying therapy. Due to this risk, cardiovascular patients should be given priority when health care systems are back to full capacity, even if other specialties have goals that conflict [4].

Whereas COVID-19 has had a worldwide effect, geographical differences exist in the pandemic’s impact. Certain locations have not witnessed a large increase in instances correlated with social and health care adaptation efforts, or the increase has disappeared and was less severe than anticipated. In some regions, there are health sector assets that can be rapidly reassigned. As areas that continue to treat the COVID-19 pandemic, there is a chance that routine heart care with the right safety measures can be slowly brought back.

The cardiovascular societies have issued a variety of policy or advisory releases that emphasize providing cardiovascular care at the pandemic’s peak [6,7,8,9,10,11,12]. These materials underscore the key concept of combining necessary cardiovascular care services while avoiding risk and preparing health care assets to fight the pandemic. As the COVID-19 epidemic subsides, it will be essential to create suitable approaches for the reintroduction of normal cardiovascular care. 

Clinics and surgeons might respond to long delays in elective procedures and clinic visits in a variety of ways, as well as by extending weekday hours or doing surgery on weekends. Delayed venous therapy may result in the progression of thrombotic illness, embolization, or worsening of postphlebitic limb morbidity. The relationship between vascular surgeons and public health professionals and epidemiologists must continue in order to examine the pandemic’s effect and the reaction to the public health problem.

## 4. The Impact of COVID-19 Infection on the Cardiovascular System

Recent findings strongly imply complicated interactions among COVID-19 and the cardiovascular system, with worse results for individuals with preexisting pathologies and the potential for significant and persistent cardiovascular impairment [13,14].

Since the discovery of SARS-CoV-2, a variety of cardiovascular consequences, such as myocardial infarction, heart failure, arrhythmias, and thromboembolic illness, have been reported [15,16]. Involved are stress caused by hypoxia, hypotension, and tachycardia; the development of acute coronary syndromes or arrhythmias; direct viral invasion; and the consequences of inflammatory responses and coagulopathies. In particular, coronary artery disease and cardiovascular risk factors, including diabetes, obesity, chronic renal disease, and hypertension, are linked to a higher risk of significant COVID-19 infection symptomatology and death [17,18]. The significance of angiotensin-converting enzyme 2 (ACE2), a constituent of the renin–angiotensin system (RAS) and the receptor by which SARS-CoV-2 promotes transmission, has been related to these connections [18,19,20,21,22].

The association among SARS-CoV2 infection and cardiovascular disease is not revolutionary, as other viruses in the Coronaviridae family, such as SARS-CoV and Middle East respiratory syndrome (MERS-CoV), have been recognized for a long time to be related to myocarditis and heart disease, potentially via ACE2 tropism [23,24,25,26,27].

### 4.1. Thromboembolic Condition

COVID-19 is linked to serious arterial and venous thrombotic events, particularly myocardial infarction, ischemic stroke, pulmonary embolism, and venous thromboembolism, especially in hospitalized patients [28,29,30]. Clinical findings in China and New York suggested that biochemical indicators of coagulation were modified in COVID-19, with nearly one hundred percent of patients with severe illness exhibiting moderate thrombocytopenia and high D-dimer values [31,32]. Subsequently, additional prothrombotic irregularities have been documented in COVID-19, notably, elevated concentrations of fibrinogen degradation products (FDPs), factor VIII, and antiphospholipid antibodies, as well as lower values of protein C, protein S, and antithrombotic proteins [33]. D-Dimer is considered an individual contributor to mortality [34,35]. Elevated D-dimer concentrations are common in severe disease and have been linked to the severity and mortality of several major viral diseases, including Ebola, influenza, HIV, and dengue [36,37]. Investigations revealed that people with COVID-19 are more likely to present with arterial and venous thrombosis compared to people with other types of viral pneumonia. This shows that pathophysiological processes have nothing to do with being immobile in the hospital [38].

Microvascular thrombi, neutrophil extracellular traps (connections of extracellular neutrophil-derived DNA), and neutrophil platelet aggregates leading to significant microvascular destruction and thrombotic obstruction are repeatedly documented in postmortem findings in several case studies [39,40]. Such vascular alterations have been partially attributed to deregulation of the endothelium’s ACE2 receptor and a prothrombotic condition [40]. An autopsy reveals extensive microthrombi in several organs, such as the heart, lungs, and brain [41]. Such processes might cause multisystem organ failure in people with severe COVID-19 variants [42].

### 4.2. Mechanisms of Coagulopathy Linked with COVID-19

Compared to non-COVID-19 pathologies, the pathophysiology of thromboembolism in COVID-19 appears to be more platelet-sensitive, with viral-mediated endothelial inflammation and hypercoagulability linked to higher levels of coagulation factors, accumulated antiphospholipid antibodies, and lowered levels of intrinsic anticoagulant proteins. Variables that significantly lead to thromboembolic illness in COVID-19 are immobilization, systemic inflammation, platelet activation, endothelial impairment, and blood circulation stagnation, all of which enhance coagulation and microvascular and macrovascular thrombosis [43]. Uncertainty remains about whether these mechanisms are exclusive to SARS-CoV-2 disease or thrombo-inflammatory consequences of severe viral illness.

In vitro investigations showed that SARS-CoV-2 disease stimulates the reactivity of platelets, neutrophils, and endothelial cells, along with modulation of coagulation factors, thrombin fabrication, fibrin generation, enhanced plasminogen activator inhibitor-1 (PAI-1) to tissue plasminogen activator (t-PA) ratio, and generation of proinflammatory cytokines, all of which encourage hypercoagulation [44]. Particularly, direct viral infection of pneumocytes and endothelial cells induces an immune and inflammatory reaction represented by activation of T cells, neutrophils, macrophages, monocytes, and platelets, resulting in cytokine generation (IL-1, IL-6, IL-10, TNF), enhanced PAI-1 expression, and thrombus formation [29]. Microthrombi in COVID-19 often include fibrin, platelets, neutrophils, and neutrophil extracellular traps (NETs), which are DNA tangles of neutrophils that have degraded [42]. NETs additionally promote hypercoagulation by engaging platelets and promoting the extrinsic pathway [45].

A publication study revealed that COVID-19 infection is related with developed hypercoagulability. COVID-19 is linked to increased thrombus development in extracranial carotid arteries, according to a study by Nypaver et al. [46].

Additionally, Ilonzo et al. [47] documented the thrombotic consequences of 21 extremely ill New York City patients at the peak of the epidemic. The cumulative death rate was 28.6%, with almost three-quarters exhibiting acute arterial thrombosis and severe ischemia. It has also been shown that individuals with COVID-19 who develop venous thromboemboli have a higher death rate [48].

### 4.3. COVID-19 and Peripheral Venous Diseases

A study conducted has shown that the coronavirus may cause thrombotic problems as a result of the cytokine storm that initiates a systemic immune reaction [49]. Consequently, infected individuals are more likely to have a hypercoagulable condition characterized by arterial and venous thrombosis. Determinants include excessive inflammation, platelet activation, endothelial dysfunction, and stasis [50]. It seems that venous thromboembolism is more prevalent than arterial thromboembolism [51]. 

In the future, vascular surgeons will need to evaluate not just a patient’s reaction to a venous treatment but also if a previous or present COVID-19 infection has affected coagulation. Unexpected concerns, such as the necessary time to appropriately execute a venous ablation on COVID-19-infected individuals who are clinically asymptomatic, remain unclear. It is becoming evident that the hypercoagulable period of COVID-19 may continue further than the acute hospitalization and into the convalescent phase; hence, anticoagulation could be required to be prolonged beyond the already recognized recommendations. 

The unknown percentage of individuals who might postpone therapy for health conditions out of worry over getting infected with COVID-19 by requesting medical care is reason for concern. In the pandemic, emergency room visits for acute cardiac events have decreased significantly, whereas at residence mortality has increased [52]. Due to the patient’s hesitation or failure to obtain early medical assistance, secondary collateral morbidity and death rates must be accounted for when calculating the health care effect of COVID-19 [53,54].

### 4.4. Cardiovascular Implications of ACE2 Being the SARS-CoV-2 Receptor

ACE2 is a carboxypeptidase that appears in soluble and membrane-bound varieties. Most of the N-terminal region, such as the catalytic region, of membrane-bound ACE2 is extracellular, whereas a transmembrane domain region sustains it on the cell membrane. The soluble version of ACE2 is broken and exported as the N-terminal ectodomain and is generally detected in tiny quantities in the bloodstream; however, it can rise in pathological circumstances [55,56,57]. Angiotensin-(1-9; Ang-(1-9)) and angiotensin-(1-7; Ang-(1-7)) are produced from Ang I and Ang II, respectively, by the catalytic action of ACE2 in the renin–angiotensin system [58]. The principal feature of ACE2 function is Ang-(1-7), which attaches to the Mas receptor and induces a vasodilator response as well as cardioprotective antiproliferative, anti-inflammatory, antifibrotic, antithrombotic, and anti-arrhythmogenic properties [59,60,61]. The series of actions resulting in the activation of the Mas receptor is known as the ACE2-Ang-(1-7)-Mas receptor axis and constitutes the defensive aspect of the renin–angiotensin pathway [61,62,63].

COVID-19 is associated with SARS-CoV2, which utilizes ACE2 as its host cell entry receptor [64,65,66]. Heart and endothelial and vascular smooth muscle cells of arteries, veins, and lymphatics express ACE2 [67,68]. The tissue dispersion of host entrance receptors is thought to generally match with viral tropisms; therefore, SARS-CoV-2 might potentially infiltrate and attack a certain tissue or region that expresses ACE2 [69,70]. Nevertheless, this theory has been contested, since ACE2-expressing enteric cells were resistant to SARS-CoV infection [71]. The same results have been observed for SARS-CoV-2, for which an indication of ACE2 expression and replicative infection in endothelial cells is lacking [72]. Various components, including transmembrane protease serine 2 and other proteases and proteins, could be required for viral infection [73,74].

Even though the major target of SARS-CoV-2 has been the respiratory tissue, evidence exists that the virus actively infects cardiomyocytes of the heart through ACE2, resulting in viral myocarditis [75,76]. In animal settings, it was shown that lung infection with SARS-CoV led to an ACE2-dependent myocardial infection, followed by a reduction in ACE2 expression [77].

In addition to its function as an entrance receptor for SARS-CoV-2, ACE2 may have an additional role in the pathogenesis of COVID-19 by modulating the inflammatory reaction. Inflammation and engagement of immune activation, which in COVID-19 might well be accompanied by a cytokine storm, are essential to many of these events [78].

### 4.5. Thromboprophylaxis for the COVID-19

Regardless of the definite correlation among hypercoagulable conditions and COVID-19, it is unknown to what degree SARS-CoV-2 increases the likelihood of thromboembolic illness. Several investigations failed to demonstrate changes in hospital-acquired venous thromboembolism between individuals with COVID-19 and non-COVID-19 sickness, indicating that the coagulopathy is not caused by the virus but rather by the disease’s overall severity and its consequences [43]. Due to the lack of restrictions, current recommendations advise that thromboprophylaxis must be explored for all admitted COVID-19 patients [28,41,79].

In a customized care strategy, prompt diagnosis and treatment of thromboembolism risk, depending on C-reactive protein or D-dimer levels, and approaching cytokine storm, depending on serum ferritin, were linked to higher COVID-19 surviving and hospital results [80].

In admitted patients with COVID-19 that do not have contraindications, including hemorrhage, the American College of Chest Physicians recommends prevention with low-molecular-weight heparin or fondaparinux instead of straight oral anticoagulants or fractionated heparin [81]. Nevertheless, optimum anticoagulation guidelines remain uncertain, and prospective clinical studies are anticipated to discover the most effective treatment options.

Infection with viruses has been associated with an elevated risk of myocardial infarction and cardiovascular risk since the early 20th century, with the largest frequency of heart disease occurring during the first 7 days after infection, according to research [81]. In contrast, in the first worldwide dissemination of COVID-19, fewer cases of acute myocardial infarction were documented than in prior years [82,83]. This decrease was mostly the result of patients’ altered behavior. Individuals were more prone to die at home or arrive late if they delayed medical interactions [84]. These behaviors might well be understood by hospital-related social fear, social separation tactics, and a decrease in regular outpatient routines [83]. Early presentation with a more complicated disease can undoubtedly increase the likelihood of persistent chronic complications.

## 5. COVID-19 Cardiovascular Symptoms

As COVID-19 is largely a disease of the respiratory system, the cardiovascular symptoms are significant in the context and therapy of all clinical risk groups for infected people. Included among them are temporary and chronic myocardial dysfunction, cardiogenic shock, arrhythmias, and vascular thrombosis. Furthermore, cardiovascular disease, hypertension, and diabetes mellitus are some of the most prevalent comorbid conditions identified among COVID-19 patients, according to multiple studies [17,85], and the observed mortality rate within patients with cardiovascular disease is 10.5%, particularly in comparison to 0.9% in patients without comorbid conditions [86].

Cardiovascular illness seems to have a significant effect on mortality if linked with myocardial damage, as individuals with an increased troponin T level had a higher incidence of malignant arrhythmia and a death rate of over 69% [87]. Virus-induced myocarditis and myocardial impairment resulting from hypoxia or cytokine-mediated systemic inflammation are suggested reasons for the cardiac dysfunction found in COVID-19 infections. As documented in the SARS-CoV-2 disease, reduced functioning of angiotensin-converting enzyme-2 in the cardiac area has also been identified as a possible modulator of cardiac dysfunction [88].

The prothrombotic condition that has been associated with SARS-CoV-2 is of special importance for cardiothoracic surgeons. In an investigation of 1099 patients from over 550 hospitals in China, 46.4% of those evaluated had increased D-dimer levels (260/560), and this finding was a potential indicator of death [89]. Microvascular thrombosis and endothelial damage of the pulmonary circulation have also been recognized as possible mediators of the substantial hypoxemia seen in severe cases.

Current anticoagulation guidelines support the use of preventive low-molecular-weight heparin in all hospital admissions without limitations [90]. Every cardiovascular procedure, including mechanical circulatory support and conventional monitoring/infusion catheters, should take thrombosis risk into account. In mechanical circulatory therapy, anticoagulation with unfractionated heparin may also be needed.

Given the poor outcome of COVID-19 patients suffering from acute coronary syndrome [91], COVID-19 infection after cardiac surgery could be linked with significant mortality, irrespective of the STS score. In 34 patients having elective noncardiac surgery in Wuhan, China during the initial stages of the pandemic, all patients became infected with COVID-19, 44 percent needed ICU hospitalization, and 20 percent died [92].

### 5.1. Ischemia and Non-Ischemia-Related Cardiac Injury

Generalized viral infection can predispose patients to acute first-type myocardial infarction owing to elevated concentrations of circulating pro-inflammatory cytokines in COVID-19 and subsequent macrophage involvement inside atherosclerotic plaques, resulting in plaque rupture and thrombosis. Acute SARS-CoV-2 infection, on the other hand, is associated with a prothrombotic and pro-coagulable condition; when combined with risk factors such as diabetes, which are associated with defective fibrinolysis and increased platelet activation, coronary thrombosis can develop [93]. The second type of myocardial infarction due to supply and demand imbalance might develop in the context of flow-limiting obstructive coronary disease, a lower blood oxygen level from COVID-19-related respiratory arrest, arrhythmia, shock syndromes, and an acid-base or electrolyte imbalance. The second type of myocardial infarction should only be diagnosed when there are signs of myocardial ischemia, such as new ischemic ECG changes, the formation of pathologic Q waves, or signs of new regional wall movement abnormalities or the destruction of viable myocardium.

### 5.2. Cardiomyopathies and Myocarditis

Cardiovascular insult is related to a background of cardiovascular illness, and increased cardiac biomarkers, in the situation of COVID-19, are linked with a worse prognosis than other types of non-acute coronary syndromes or myocardial damage [93]. There have been reports of direct effects of viral invasion on the heart, although they seem to be rare [94]. Cardiac pericytes and cardiomyocytes produce ACE2 transmembrane receptor proteins, and it is hypothesized that their combination with the S protein of SARS-CoV-2 coronavirus is the cause of cell invasion [95]. To date, though, postmortem findings of direct cardiac involvement in dead individuals give an indication of viral particles in the interstitial fluid or macrophages instead of in cardiomyocytes [96].

Patients with COVID-19 have shown lymphocytic or eosinophilic infiltration of the heart, whether at autopsy or upon endomyocardial biopsy. The preponderance of findings are case studies or small-sample series, which makes it difficult to determine the prevalence of myocarditis in admitted COVID-19 patients [97]. Myocarditis in individuals with COVID-19 has been found to be due to cytokine-induced inflammatory myocarditis, instead of being a consequence of viral invasion [97].

Severe abnormalities in systemic microvascular and endothelial activity have been described in COVID-19 patients, especially those needing mechanical ventilation [98]. After an infection, coronary microvascular and endothelial dysfunction are probable sources of cardiac injury and chronic symptoms. Additionally, there have been instances of diffuse systemic vasculitis and endothelial dysfunction that might make a substantial contribution to decreased microvascular activity in COVID-19 individuals [99,100,101]. Postmortem, microvascular thrombosis has been seen in lung parenchyma, and this might be a source of microvascular destruction or malfunction in the coronary arteries [102]. The outcome of mechanistic image investigations assessing coronary microvascular circulation and myographic vascular function in individuals with concurrent coronary evaluation after SARS-CoV-2 infection is expected [103].

Takotsubo cardiomyopathy is a recognized acute cardiotoxic consequence of COVID-19 infection that might be caused by direct myocardial damage, inflammation, and stress [104,105]. Catecholamine influx and cytokine storm are mechanisms involved in COVID-19-associated cardiomyopathy.

### 5.3. Arrhythmia and Abrupt Cardiac Death

Atrial fibrillation is the most common arrhythmia initially detected in COVID-19 patients, appearing in about one-fifth of admitted patients and being related to an elevated risk of death [106]. Premature ventricular complexes, ventricular tachycardia, and bradycardia have also been documented; however, it should be emphasized that a percentage of these individuals had previous rhythm problems [107]. In published research, acute cardiac mortality was observed; nonetheless, these patients were taking QTc-prolonging drugs, such as quinolone antibiotics or hydroxychloroquine, that have proven to be minimally effective in admitted patients [108].

### 5.4. Heart Failure

Infection with COVID-19 might increase susceptibility to abrupt heart failure due to the revelation of subclinical underlying heart failure or direct cardiac dysfunction. Virus-induced invasion of inflammatory cells, proinflammatory cytokines, endothelial damage, microthrombosis, and hypoxia due to respiratory failure are the processes behind SARS-CoV-2-related heart failure [109]. Once admitted to a hospital, COVID-19 patients are at an elevated risk of heart failure with maintained ejection fraction, which is linked with cardiac anatomical and functional irregularities, myocardial damage, and the possibility of long-term heart disease [110]. Consequently, comprehensive monitoring and cardiac evaluations, involving echocardiographic measurement of left ventricular diastolic function and cardiac markers, have been recommended for regular management of COVID-19 patients [111].

Due to immunosuppression and hemodynamic instability, heart failure patients on complex therapy, particularly those needing a heart transplant, require specialist treatment and the participation of advanced heart failure team members, since they are at a really elevated risk [112]. In moderate to grave cases of COVID-19, the recommendations of the International Society for Heart and Lung Transplantation include delaying immunosuppressive medicines [113]. There have been reports of effective heart transplantation for COVID-19-associated post-infectious fulminant myocarditis [114]. Different cardiovascular conditions associated with COVID-19, mechanisms implicated, and therapy/management used are shown in Table 1.

## 6. Solutions

### 6.1. Management of Cardiovascular Disease in COVID-19-Exposed Patients

Choices concerning the therapy of COVID-19 individuals with prior cardiovascular disease must be made on an individual basis. Cardiovascular patients should be kept as far away from people infected with SARS-CoV-2 as possible. Unless contraindicated, vaccination against SARS-CoV-2 should be recommended in cardiovascular patients [4,115,116,117].

### 6.2. The Possibility of Using Renin–Angiotensin System Antagonists as a Treatment in COVID-19 Pneumonia

The intensity of acute lung collapse could be lessened by inhibiting the renin–angiotensin system. SARS-CoV infections can be prevented by ACE2 antibodies and soluble ACE2 molecules [118]. IgG1 Fc-fused recombinant ACE2 proteins have been shown to have significant in vitro neutralizing action towards SARS-CoV and COVID-19 [119]. By inhibiting renal and cardiac fibrosis, xanthenone, an ACE2 activator, has been demonstrated to lower blood pressure and enhance heart activity in cases of spontaneous hypertension [120]. It is possible that enhanced ACE2 function is a compensatory response to hypertension, given that circulating ACE2 function elevates with rising vascular tone [121]. Recombinant human ACE2 has been demonstrated to have positive cardiac benefits in a similar manner [120,122]. It has been shown that the ACE inhibitor/Angiotensin II receptor blocker antagonist renin–angiotensin–aldosterone system reduces inflammation in COVID-19 pneumonia, hence reducing mortality [123,124]. In a retrospective study, however, there was no connection among severe COVID-19 outcomes and chronic ACE inhibitor/Angiotensin II receptor blocker medication [125]. In a subgroup analysis of diabetic individuals on a renin–angiotensin system inhibitor, de Abajo et al. [126] found that the probability of COVID-19-related hospitalization was decreased in those patients. In individuals with hypertension, use of a renin–angiotensin system inhibitor was not linked to higher mortality or severity of COVID-19, according to a comprehensive review and meta-analysis. Nevertheless, the Angiotensin II receptor blocker was linked to decreased mortality [127]. In another investigation, there was no indication of increased severity of COVID-19 illness in London patients receiving chronic ACE inhibitor or Angiotensin II receptor blocker therapy [128]. In contrast, severe COVID-19 patients persistently treated with an ACE inhibitor/Angiotensin II receptor blocker had a higher risk of acute renal damage compared to the 149 individuals from the Referral Center Cohort in the northeast of France [129].

### 6.3. Additional COVID-19 Therapy Suggestions

COVID-19 infection must be treated in order to limit the progress of the illness, namely pneumonia [130,131,132]. Furthermore, if individuals also have cardiovascular disease, they must be managed promptly. In light of previous clinical trials of SARS-CoV-2 and Middle East respiratory syndrome, epidemiological features, clinical phenotype, and treatment results of COVID-19 patients with concomitant cardiovascular disease, the therapy outlined following is advised [130].

General and symptomatic therapies, antiviral medication, hypoxia and dyspnea therapeutic interventions (oxygen therapy, noninvasive and invasive ventilatory assistance), circulatory support treatment for patients in shock, prompt utilization of antibiotic agents under proof of secondary infection, therapeutic interventions for cytokine storm, and glucocorticoid medication in severely ill patients [131] are recommended. It is recommended to provide oxygen to hypoxia patients using nasal prongs, face-mask, high-flow nasal cannula, or noninvasive ventilator. Consequently, extracorporeal membrane oxygen assistance and artificial breathing are required. Additionally, renal replacement treatment may be required in certain situations [131]. Seventy-five percent of patients have undergone antiviral therapy, notably ganciclovir, lopinavir–ritonavir, and oseltamivir [132]. In one research study of 138 COVID-19 patients, antiviral medications were delivered to 89.9% of patients [133]. Even a broad-spectrum antiRNA drug used to treat Ebola has been tested for COVID-19 [134]. Significantly, it is advised that the amount of clotting time tests be augmented in atrial fibrillation patients receiving COVID-19 medication in conjunction with oral anticoagulants [129].

Additional drugs/agents that have been explored are arbidol (umifenovir; mostly used in Russia and China, but not Food and Drug Administration-approved for use in the United States), interferons, intravenous immunoglobulin, chloroquine, and plasma taken from patients that recovered from COVID-19 [130]. Furthermore, traditional Chinese herbs have been assessed by Chinese medical practitioners [130]. Antibiotics and antifungals are also recommended when co-infections are confirmed or even suspected [130]. Moreover, chloroquine possesses a promising suppressive effect. However, therapeutic use of chloroquine can lead to very adverse effects. In particular, hydroxy-chloroquine has an anti-viral effect that is extremely comparable to that of chloroquine and may, thus, be utilized as a more suitable therapeutic method. In particular, hydroxy-chloroquine may reduce the severe increase in COVID-19, by controlling the cytokine storm through the regulation of T-cell activation. Consequently, this drug’s therapeutic character is safer, making it ideal for pregnant women [135].

In a meta-analysis, Sarma et al. indicated that hydroxy-chloroquine might be promising for reducing the number of patients with radiological progression, with a safety profile equivalent to that of standard therapies [136]. Chowdhury found that five of seven studies including chloroquine or hydroxychloroquine resulted in good results for patients, whereas two of seven trials showed no improvement in comparison to controls [137]. Nevertheless, a meta-analysis by Singh et al. revealed that there is no advantage in viral clearance, although a substantial increase in mortality was seen in three studies (n = 474) with hydroxychloroquine in patients with COVID-19 [138].

Several medicines, such as chloroquine and hydroxychloroquine, might contribute to the progression of malignant arrhythmias by prolonging the QT interval [139,140,141]. In addition, treatment with hydroxy-chloroquine and chloroquine with medications that suppress the CYP3A4 enzyme may increase the risk of QT interval lengthening [139,140,141,142]. Consequently, individuals with arrhythmias should be under constant ECG monitoring while receiving supportive therapy [142,143,144,145,146,147].

Clinicians must therefore delay the routine use of chloroquine and hydroxy-chloroquine until the results of more thorough studies with more precise parameters become more definite [142].

### 6.4. Transplantation of the Heart with Mechanical Circulatory Support

Multiple problems make the usual operation of a heart transplantation program problematic or unfeasible, especially in areas with high COVID disease incidence [148]. Newly transplanted patients who need immunosuppression are likely at significant risk for COVID infection, especially in hospitals located in severely contaminated regions. A report from China says that COVID infection was found in two orthotopic heart transplant patients [148]. One of them had few symptoms and recovered normally, while the other had to be hospitalized, take antiviral medicine, and temporarily stop immunosuppression.

The Extracorporeal Life Support Organization continues to recommend veno-arterial extracorporeal membrane oxygenation (VA-ECMO) for basic indications; however, disclaimers for COVID-19 patients remain as to the choice of health care organizations based on work-related risk and expenditure limitations [149].

Implantation of a left ventricular assist device (LVAD) presents many complications in the context of COVID-19. Such individuals frequently experience a severe inflammatory reaction, and the COVID-19 infection might worsen. Individuals with LVADs have elevated interleukin-6 cytokine levels six weeks after implant [150,151], that are believed to correspond to the severity of COVID-19 [152].

Moreover, due to the now-described COVID-19-related coagulopathy, infected patients with LVADs might be challenging to anticoagulate, hence, increasing their risk of thrombotic problems. Lastly, if a patient with an LVAD had acute respiratory distress syndrome, it would be difficult to perform prone ventilation due to the possibility of outflow graft and driveline compression and lower cardiac output because of reduced venous return [153,154]. In conclusion, throughout the COVID-19 pandemic, it would be preferable to postpone this procedure as far as feasible due to its extreme physiological impact and resource complexity. Measures to be taken into consideration when performing a surgical procedure are shown in Table 2.

## 7. Vascular Surgeons and Ethics-Related Factors

Prior to the COVID-19 pandemic, doctors treating vascular disease routinely conducted a range of elective, urgent, and emergency interventions. When combined with cases of nonselective arterial disease, urgent and emergency cases of venous thromboembolism and vascular trauma make up 30 to 50 percent of the different types of cases seen in a busy vascular surgery practice [155].

Past studies have shown a high risk of morbidity and death in patients identified with COVID-19 who need elective or emergency surgery, independent of preoperative or postoperative COVID-19 contraction. Thoracic surgery patients were a minority of this previously published cohort, and data related to adult cardiac surgery patients with COVID-19 are restricted to case studies, with very few large-scale findings [156,157,158].

This established order was disturbed by the coronavirus epidemic, endangering the timeliness and effectiveness of care (Figure 1). By bringing patients into a hospital or office environment, physicians are faced with the problem of possible COVID-19 contact with patients. This prompted clinicians to reconsider possible exposure and hospital resource consumption for COVID-19-associated admissions [159]. According to a worldwide study conducted by Ng et al. [160], 86.9% of vascular surgeons have discontinued or reduced their outpatient services in reaction to the pandemic.

Vascular surgeons’ practical work had to evolve away from desired face-to-face encounters and embrace a “just if your existence relies on it” approach for contact with patients directly due to a lack of preparedness. The evolution of postoperative follow-up treatment and chronic illness monitoring has been accelerated by “remote” medicine. Telehealth through phone conversations, video chats, and the digital medical record are illustrations of such methods. This quick shift in vascular practice management has only been made feasible by technological and internet advancements.

For patients who could not engage in telemedicine consultations through any of these networks, other alternatives, including mobile video chat programs or even voice conversations, were made accessible. A tiny percentage of patients, including those needing physical inspection or assistance, were not suitable for virtual visits and were instead visited in person. Despite the particular criteria of emergent and elective surgery during a time of high risk of COVID-19 infection, it is crucial that surgeons actively follow postponed patients. These patients are at risk for disease progression as they wait, particularly because they lack usual contact with their cardiologists, general care physicians, and pharmacists.

Guidelines and methods of practicing medicine, particularly cardio-vascular surgery, have changed throughout the pandemic period and continue to do so, with some modifications including optimization of therapy principles and, in particular, a complete verification of the benefits and contraindications of the procedure optimized for each patient’s condition. 

Appropriate values to be operationalized include the following [161]: (1) optimizing outcomes in order to save the greatest number of lives or life years possible by prioritizing operations or examinations that will benefit more people and to a greater extent than those that will benefit fewer people to a lesser extent; (2) equality, such that similar cases are addressed similarly while accounting for guideline health disparities; and (3) reasonableness, ensuring that the risk of further harm is minimized. Lastly, procedural justice needs to be supported by an ethical framework [162] to make sure that all decisions are based on the best information available and are shared openly.

## 8. Transition of Cardiovascular Surgery

Despite the possibility of a reduction in the prevalence of COVID-19, the burden on hospital systems is prone to be experienced for an extended amount of time. In light of this, it is probable that the most negatively affected hospitals may not be prepared to start elective surgeries soon, or at least not immediately, at their previous levels. As the proportion of admitted coronavirus patients reduces, it is probable that cardiothoracic surgical workload will gradually return, with the sickest patients receiving priority for early surgery. As operational activity gradually increases, a considerable backlog of patients whose treatment was postponed owing to the COVID-19 outbreak is anticipated to develop. Therefore, it is necessary to keep managing the “postponement list”, incorporating communication with patients and their doctors, so that patients who are suitable for surgery may be appointed when capability resumes. Researchers are also encouraging the referring doctors to discuss the people they are not sending because of the crisis so that surgeons can plan virtual appointments and construct a plan for patients who require just a quick consultation or preoperative tests to be surgically prepared. Throughout this healing period, it could be important to arrange appointments on the weekend to accommodate all of these patients. In summary, the pandemic had a significant impact on the vascular sector; vascular surgeons had to adjust their everyday practice and react to the situation by reallocating resources.

In Portugal [163], the percentage of visits to the emergency department for vascular surgery decreased by 43.3% in the first 4 weeks following the initial diagnosis of COVID-19. During the pandemic, patients came to the emergency department with more serious and progressive conditions, resulting in an increase in hospitalization and emergency/urgent surgery rates. Concern of infection and movement restrictions imposed by the quarantine might have discouraged patients from visiting the emergency department and postponed the delivery of adequate treatment.

In Italy, not infrequently compelled by the patient’s wish to avoid surgeries during this interval, surgeons have tended to remove low-value therapies and adjust therapeutic strategies [164,165,166]. For a considerable number of patients with cardiovascular diseases, it is difficult to determine which procedures may be postponed. Interventions thought to have a minimal benefit were frequently postponed or even discontinued, but individuals with vascular disease considered this choice challenging. In Rome’s four main hospitals, surgical and elective outpatient admissions decreased at a pace comparable to that of Pavia. In 2020, the total number of vascular operations performed at the Hospital San Matteo was lower than in prior years (625 versus 961) [167]. They stopped procedures for asymptotic carotid occlusive disease, endovascular procedures for claudication, and minor abdominal aortic aneurysms. There was a rise in the number of procedures for acute ischemia of the lower limbs due to emboli and abdominal aortic aneurysm rupture. COVID-19 infection and death are more likely to occur in patients with cardiovascular disease who are over the age of 65 and who have pulmonary disorders [168,169]. The most prevalent risk variables for post-operative lung infection and pulmonary significant events are general anesthesia with endotracheal intubation, postoperative discomfort, and ICU permanence. Endovascular surgery and other minimally invasive surgical methods often involve a reduced operation time, no general anesthetic or tracheal intubation, and less administrative effort. The length of hospital stays has decreased. The incidence of emergency visits in Pavia for ruptured abdominal aortic aneurysm and acute ischemia of the lower extremities increased in 2020 compared to the preceding year. This issue was not apparent in Lazio, in which the pandemic was less severe [170]. During the COVID-19 pandemic in Lombardy, the most common vascular illness needing surgical treatment was acute limb ischemia. COVID-19 was linked to a fourfold increase in the risk of mortality and a threefold rise in the risk of serious undesirable consequences [171]. This comprehensive experience revealed that acute limb ischemia was the most prevalent vascular illness needing hospitalization and surgical treatment, particularly in COVID patients. Prior global reports have shown the connection among SARS-CoV-2 infection and microvascular inflammation [135], distal vasculitis, and the prothrombotic condition [171]. These findings appeared to be associated with the inflammatory cytokine storm (interleukin-6 and interleukin-1 beta) that contributes to the procoagulant and proadhesive condition of the defective endothelium [171]. In addition, inadequate coagulation measures are often correlated with a poor prognosis in COVID-19 patients [172,173]. This could be the reason for the results of Bellosta et al. [174], who initially documented a rise in the frequency and extent of native peripheral artery obstruction in COVID-19 patients. This might be connected to the increased hypercoagulability linked with a D-dimer elevation, which was seen in COVID-19 patients [175,176].

In a Singapore tertiary hospital’s vascular surgery section, medical capacity was reduced by 50 percent during the pandemic, but a fast-access diabetic foot clinic continued to operate, since diabetic foot patients often need more rapid and more urgent care [177]. Every COVID-19 probable and confirmed positive patient procedure was performed in a surgery room with negative pressure. An insightful conclusion is that a small minority of COVID-19 patients may have a lengthy and challenging ICU hospitalization, necessitating extended breathing and inotropic support. This may cause peripheral vasoconstriction and gangrene in the upper and lower extremities. One of their COVID-19-positive patients experienced peripheral gangrene of all four limbs due to noradrenaline and dopamine overuse at high doses. The patient ultimately died from COVID-19 complications.

During the COVID-19 pandemic, the practice of vascular surgical treatment in the United States was drastically altered, and local differences in clinical management were observed [174]. These included substantial cancellations of elective surgical procedures, a decline in outpatient visits, and decreased use of the vascular laboratory. Telemedicine was used to maintain vascular therapy for patients who prioritized life over limb.

The economic effect of the COVID-19 epidemic has drastically altered the United States’ healthcare system. During the pandemic, a multiplicity of de novo stresses exacerbated the general nurse deficit, exposing and aggravating long-term difficulties that have hampered the nursing profession for years. In the midst of drastically increasing nurse labor costs and continuous clinical income production, the studies [178,179] reveal a serious degradation of the profit margin in vascular surgery. As the country fights to recover from the pandemic, it is probable that the expenditures in nursing education, training, and recruiting should be made permanent.

### Effects Worldwide

Due to the limitations or patients’ unwillingness to access health care services at the peak of the pandemic, there was also an increase in late presentations of prevalent cardiac diseases. There might be an increase in the incidence of late sequelae of myocardial infarction, including heart failure and postinfarct ventricular septal defect. Individuals with valvular disease might exhibit signs of severe or decompensated heart failure, as well as diminished ventricular activity. 

Anecdotally, the reported frequency of aortic dissection presentations for acute treatment has decreased, and chronic dissections may occur as a result. There is proof that individuals are seeking medical attention less commonly; in a study, more than half of participants described a 40% to 60% decrease in myocardial infarction hospitalizations [180]. Eventually, this delay in treatment might raise the prevalence of chronic cardiac disease and lead to the need for more extensive follow-up care and rehospitalization, thus increasing pressure on a hospital system that is recuperating.

As humans proceed to face the COVID-19 pandemic, it is evident that cardiac surgical initiatives will necessitate multidisciplinary collaborative efforts, an eagerness and versatility to change institutional practices, and the capacity to preserve core surgical skills while applying them to the care of critically ill patients. The effect of every initiative will depend on the equilibrium between the severity of COVID-19 infection and resource capabilities, and a reversion to normality is expected after the spike and maximum number of cases have passed.

## 9. Residency and COVID-19

Within the first pandemic wave, vascular surgery operations decreased by an average of 23.3%, and 53.5% of vascular surgery residents were reassigned to other specialties [181].

According to three out of four respondents, the postponement or cancellation of many outpatient visits and elective procedures caused a significant reduction in surgical operations for vascular trainees and other vascular practices, which negatively impacted their residency despite the fact that they were not on the COVID-19 front lines.

To assess the status of vascular learning throughout Europe during the first wave of the COVID-19 pandemic, in the study of Pereira-Neves et al. [182], an electronic questionnaire was sent to 104 vascular trainees in 27 European countries. In total, 73.5 percent of respondents said that the COVID-19 pandemic had a detrimental influence on vascular education, and 73.4 percent felt that adjustment actions must be adopted.

During the COVID-19 pandemic, Iborra et al. [183], conducted six online training classes on open vascular surgery procedures for vascular trainees. A practice kit including the necessary materials for the activity was created and sent to the participants. The carotid endarterectomy was closed with a bovine patch and the anastomosis was performed using a biological prosthetic bypass. Participants were instructed to film the carotid patch closing and bypass anastomosis with their arms and the model visible; the films were further analyzed, and the results were sent to the participants through email. All the participants reported that it was a useful tool and that they would repeat the course again, being a feasible way to continue the study in a pandemic period. 

It was anticipated that the influence of COVID-19 on surgical practices in the United States would have a substantial effect on the training conditions and particular pressures for trainees [184]. Numerous modifications were made to the vascular surgeon trainees’ experiences, varying from the nature of the job requirements (redeployment to responsibilities other than those performed by a vascular surgeon’s trainee) to the volume of work requests (extensive clinical commitments and time dedicated to self-learning) or the combination of the two.

In conclusion, the COVID-19 pandemic had a considerable detrimental influence on vascular education. In light of the protracted pandemic scenario, it is felt that compensating measures must be considered in order to ensure high levels of vascular education and create new training instruments for future trainees.

## 10. Short-Term and Long-Term Consequences of the COVID-19 Pandemic Affecting Surgical Care Provision

### 10.1. The Pandemic’s Effect on Surgical Services

The pandemic’s consequences for surgery were severe, possibly long-lasting, and broad, according to published studies and growing real-life experiences. Studies [185,186,187] found in connection to operation and perioperative care included comments, anecdotal examples, and suggestions.

Certain general concepts are consistent throughout areas and could be utilized to alleviate the short- and long-term consequences of the pandemic on surgical services, as well as to inform future preparation efforts.

### 10.2. Reconstructing Surgical Capability following Pandemic

It is unknown what influence this reduction in surgical ability will have on the surgical condition and related health of patients, as well as their well-being, functional capacity, risk of function loss, and detrimental impacts on prognosis. Patients may experience sadness, disappointment, anger, irritation, and stress when their appointments are canceled under regular conditions. This is in addition to the possible economic repercussions (loss of employment, sick leave, or difficulty in sustaining tenancy) and family life impacts [188].

Ideally, the COVID-19 period is going to be a once-in-a-lifetime event for several health care professionals worldwide. This unique period of pandemic crisis necessitates adjustments and increased flexibility in cardiothoracic surgery training expectations in order to address trainee-specific and surgical education-specific concerns. Constant reorganization of the schedule and curriculum for surgical trainees is unavoidable.

### 10.3. Getting Ready for the Future

We must begin planning for the future. Telemedicine and remote monitoring technologies, which have thrived during COVID-19, ought to be incorporated into the future of practice [189]. When COVID-19 fades, we will have a significant backlog of cases. The principle of sensible prioritization also remains, such that the most important situations are handled immediately. As we expand our outpatient and inpatient care to pre-COVID-19 levels, the constant screening for patients who may be infected with COVID-19 should remain consistent, along with maintaining a high standard of cleanliness. 

Patients with limb ischemia are a frail and sensitive population whose prognosis is significantly impacted by a delay in the standard of therapy [190]. Throughout a pandemic period, it is crucial that these individuals continue to receive routine medical treatment, with caregivers recognizing them and recommending them in time for revascularization. The higher incidence of major limb amputations seen in research might have a significant influence on functional outcome and quality of life and should, thus, be avoided [191]. Patients should not be frightened to contact their physician, despite the fact that social isolation is necessary for preventing the transmission of the SARS-CoV-2 virus. In these individuals, an accurate evaluation of wounds and tissue loss must be conducted by high-quality computerized video or picture consultation or physical examination.

## 11. Conclusions

While there is a definite link between cardiovascular disease and COVID-19, it is important to note that the majority of the studies are retrospective and, therefore, susceptible to bias and interference. It is difficult to differentiate the very significant influence of age on COVID-19 results from those of other diseases whose incidence increases with age, such as hypertension, diabetes, and other cardiovascular disorders. Nonetheless, addressing this connection is crucial, since it has ramifications for patient assessment. Similarly, despite the fact that heart damage appears to be widespread in people with severe COVID-19, the long-term health consequences and any subsequent impact of cardiovascular disease are unknown.

There is currently no cure for COVID-19, although the effective use of medications including remdesivir and dexamethasone and the implementation of effective vaccination programs would certainly make a difference. As varieties of SARS-CoV-2 develop for which existing vaccinations provide less resistance, it is more probable that the pandemic will persist in a certain form for many years, with COVID-19 cardiovascular complications continuing to be a clinical issue. This possibility has prompted urgent recommendations for enhanced research into relevant pathways and better cardiovascular and cardiometabolic disease prevention and management for patients.

The worldwide pandemic brought on by SARS-CoV-2 compelled the vascular profession to reevaluate the therapy of selected vascular diseases and discover ways to address the vascular implications of COVID-19 infection. In the lack of published research or empirical evidence on COVID-19, vascular surgeons had to determine the risk–benefit of both regular and emergency treatment in the light of community-based disease incidence and facility capabilities.

## Figures and Tables

**Figure 1 medicina-58-01554-f001:**
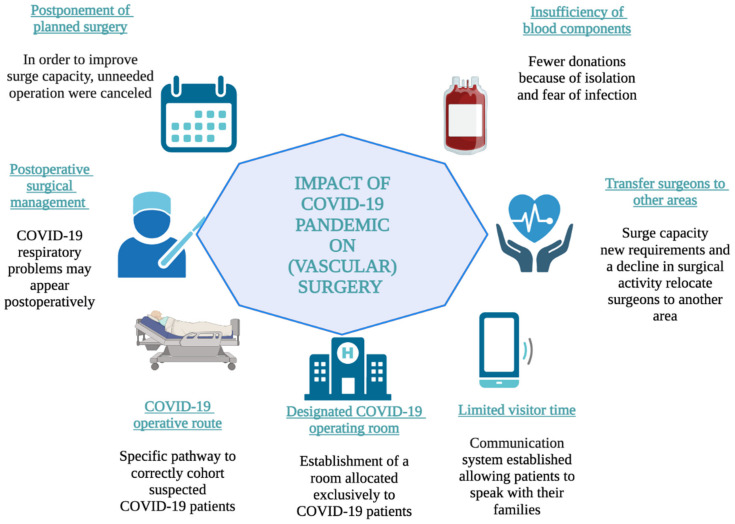
The impact of COVID-19 pandemic on (vascular) surgery.

**Table 1 medicina-58-01554-t001:** Different cardiovascular conditions associated with COVID-19, mechanisms implicated, and therapy/management used.

Cardiovascular Condition Associated with COVID-19	Mechanisms Implicated	Therapy/Management	References
Coagulopathy/thrombosis linked with COVID-19	Reactivity of platelets, with viral-mediated endothelial inflammation and hypercoagulability, higher levels of coagulation factors, accumulated antiphospholipid antibodies, and lowered levels of intrinsic anticoagulant proteins	Antiaggregant/anticoagulant therapy	[42,43,44,45,46,47,48]
Peripheral venous diseases	Cytokine storm, systemic immune reaction, hypercoagulable condition, excessive inflammation, platelet activation, endothelial dysfunction, and stasis	Antiaggregant/anticoagulant therapy, venous ablation	[49,50,51,52]
Thromboembolism	Disease’s overall severity and its consequences	Low-molecular-weight heparin or fondaparinux instead of straight oral anticoagulants or fractionated heparin	[81]
Virus-induced myocarditis and myocardial impairment	Hypoxia or cytokine-mediated systemic inflammation	Depending on the symptomatology	[88]
Myocardial infarction	Elevated concentrations of circulating pro-inflammatory cytokines, macrophage involvement inside atherosclerotic plaques, plaque rupture, and thrombosis	Antiaggregant/anticoagulant therapy	[93]
Cardiomyopathies and myocarditis	Lymphocytic or eosinophilic infiltration of the heart, cytokine-induced inflammatory myocarditis	Depending on the symptomatology	[97]
Takotsubo cardiomyopathy	Direct myocardial damage, inflammation, and stress	Depending on the symptomatology	[104,105]
Arrhythmia and abrupt cardiac death	Systemic illness, systemic infection, inflammation, QTc-prolonging drugs, i.e., quinolone antibiotics or hydroxychloroquine	Pharmacological management, i.e., amiodarone and diltiazem	[107,108]
Heart failure	Invasion of inflammatory cells, proinflammatory cytokines, endothelial damage, microthrombosis, and hypoxia due to respiratory failure	Pharmacological management	[110,111,112,113,114]

**Table 2 medicina-58-01554-t002:** Measures to be taken into consideration when performing a surgical procedure.

Preoperative	Intraoperative	Postoperative
Constant COVID-19 infection screening prior to 24 to 48 h	Non-COVID surgical suites	Clean recuperation area
History of travel and possible exposure	Adaptation of negative-pressure ORs	Considering COVID-19 infection in the event of persistent respiratory distress
Accurate CXR evaluation	Airborne measures and PPE must be used by all suppliers	Reduce the risk of renal failure and persistent respiratory distress
Avoid unnecessary testing wherever feasible	Non-essential personnel to be absent from room	Protocol for accelerated recuperation, if applicable: Extubation, mobility, and disconnection of chest tubes and pacing wires as quickly is possible
Consider using previous test results wherever feasible	Just attending-level medical personnel	Shortly following extubation, patients receive surgical masks
Presurgical care delivered through telehealth	Video laryngoscopy to facilitate intubation	Timely collaboration with the family for postoperative recuperation at residence
Patients wear surgical masks before arriving at the hospital	Reduce worker turnover inside the room	Early release when clinically stable
Prompt intubation before surgery for patients with COVID-19 and respiratory dysfunction	Whenever possible, avoid TEE to prevent pleural invasion and lung harm	Following discharge, frequent and regular digital follow-up
Putting off a case if COVID-19 positive	Limit operations that involve CO_2_ insufflation	Test for COVID-19 if clinical signs are seen

COVID-19, Coronavirus disease 2019; CXR, chest X-ray, OR, operating room; PPE, personal protection equipment, TEE, transesophageal echocardiography.

## Data Availability

Not applicable.

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
