# Peer review of "Narrative Review of New Insight into the Influence of the COVID-19 Pandemic on Cardiovascular Care"

_medicina, 2022, doi:10.3390/medicina58111554_

Round 1

Reviewer 1 Report

1. Authors could provide a table to summarize studies relating different  cardiovascular conditions with COVID-19, mechanisms implicated and therapy/management used.

2. The first paragraph in the  section "solutions" it does not really fit as TAVR and mitraclip are well recognized therapies but there is no discussion regarding COVID-19

3.The section "solutions" should include therapy managements related to COVID-19

4.In the section "Transition of cardiovascular surgery" Is there any reason to report only the impact in Italy, Portugal, Singapore an USA?, it seems a random choice and the structure of each subsection lacks consistency. Each subsection should be focused and structured in a similar way.

Author Response

Dear Reviewer,

Thank you for your suggestions and requests. We hope that we have updated the article in a satisfactory manner.

Reviewer 1

  1. Authors could provide a table to summarize studies relating different cardiovascular conditions with COVID-19, mechanisms implicated and therapy/management used.

Done. The table can be found at line 379.

  1. The first paragraph in the section "solutions" it does not really fit as TAVR and mitraclip are well recognized therapies but there is no discussion regarding COVID-19.

We have modified the content.

3.The section "solutions" should include therapy managements related to COVID-19.

Done. We have added the information [line 384].

4.In the section "Transition of cardiovascular surgery" Is there any reason to report only the impact in Italy, Portugal, Singapore and USA?, it seems a random choice and the structure of each subsection lacks consistency. Each subsection should be focused and structured in a similar way.

We have chosen the areas that have reported the most significant situations regarding our subject. The section was modified accordingly. Thank you!

Reviewer 2 Report

The aim of the paper was to perform a literature review on the effects of COVID-19 on cardiothoracic and vascular surgery care and departments. This is a useful topic as it helps improve cardiothoracic and vascular surgery care in light of the COVID-19 pandemic. The review is clear, comprehensive, well-written and relevance to the field.

General comments:

1. Introduction: Is this a structured review? I ask because I noticed the methods (lines 45-61) have been included in the introduction, instead of being a stand-alone section. I expected to see a similar structure as in the abstract.

A few of the cited references, though relevant (#58, 120, 121), are outdated (i.e., >19 years old). Are authors able to replace these with most recent ones?

Author Response

Dear Reviewer,

Thank you for your suggestions and requests. We hope that we have updated the article in a satisfactory manner.

Reviewer 2

The aim of the paper was to perform a literature review on the effects of COVID-19 on cardiothoracic and vascular surgery care and departments. This is a useful topic as it helps improve cardiothoracic and vascular surgery care in light of the COVID-19 pandemic. The review is clear, comprehensive, well-written and relevance to the field.

General comments:

  1. Introduction: Is this a structured review? I ask because I noticed the methods (lines 45-61) have been included in the introduction, instead of being a stand-alone section. I expected to see a similar structure as in the abstract.

Materials & Methods Chapter has been added. [line 46] Thank you!

A few of the cited references, though relevant (#58, 120, 121), are outdated (i.e., >19 years old). Are authors able to replace these with most recent ones?

The references were replaced. Thank you!
